# Moving from Principles to Practice: A Scoping Review of Value-Based Healthcare (VBHC) Implementation Strategies

**DOI:** 10.3390/healthcare12232457

**Published:** 2024-12-05

**Authors:** Egidio de Mattia, Carmen Angioletti, Melissa D’Agostino, Filippo Paoletti, Antonio Giulio de Belvis

**Affiliations:** 1Faculty of Economics, Università Cattolica del Sacro Cuore, 00168 Rome, Italy; antonio.debelvis@policlinicogemelli.it; 2Critical Pathways and Evaluation Outcome Unit, Fondazione Policlinico Universitario “A. Gemelli”—IRCCS, 00168 Rome, Italy; melissa.dagostino01@icatt.it (M.D.); filippopaoletti.an@gmail.com (F.P.); 3Management and Healthcare Laboratory, Institute of Management, Sant’Anna School of Advanced Studies, 56127 Pisa, Italy; carmen.angioletti92@gmail.com

**Keywords:** value-based healthcare, hospital management, operational strategies

## Abstract

Background/Objectives: The principles of value-based healthcare (VBHC) have received widespread endorsement, leading healthcare organizations worldwide to shift their strategies towards them. However, despite growing recognition and acceptance, the actual implementation of value-based approaches varies widely. This research aims to identify studies that address the implementation of VBHC at different levels (healthcare policymakers, hospital administrators, and healthcare providers), focusing on each level’s relative strategies. Methods: To this end, a scoping review was conducted in accordance with the PRISMA extension for the scoping reviews checklist. The electronic databases of Web of Science, PubMed, MEDLINE, and Scopus were searched to identify relevant publications in English from January 2006 to 31 July 2023. Results: We identified 30 eligible studies. Findings are organized into four main macro strategic levels, each comprising specific dimensions and operational approaches. Fourteen articles analyzed the role of government commitment in VBHC implementation, while six articles focused on regional integrated care systems. The role of hospitals was described in sixteen records. Conclusions: Our study suggests that a comprehensive approach is necessary for the successful implementation of VBHC. Hospitals emerge as pivotal in this shift, requiring organizational and attitudinal changes among healthcare professionals. However, a complete transition towards VBHC that ensures seamless patient management throughout the entire care delivery value chain necessitates government involvement in terms of state legislation, reimbursement methods, and hospital networking.

## 1. Introduction

The escalating concern about the sustainability of healthcare systems, driven by the ever-increasing demand for healthcare services and the challenges posed by resource limitations, has mandated a profound shift in priorities. This calls for innovative approaches to healthcare delivery, as more than mere cost containment is needed to address the mounting pressure on healthcare systems, particularly where costs are not aligned with outcomes [1]. In response, value-based healthcare (VBHC), a model introduced by Porter and Teisberg [2], represents a paradigm shift in healthcare management. Unlike models prioritizing service volume, VBHC is a tool used to enhance health organizations’ reorientation towards person-centered care [3]. A patient-centered approach necessitates a departure from the conventional discipline-based organizational model by aligning hospital activities around the needs of care processes (and patients) [4]. This approach puts the patient at the center of the care process, facilitating their active involvement in decisions concerning their health through the Shared Decision Making (SDM) approach. This method integrates clinical expertise with the patient’s values, needs, and preferences, fostering an empathetic and collaborative dynamic that strengthens the physician–patient relationship, yielding benefits for both parties [5].

Since their introduction, the principles underlying VBHC have proven to be compelling and widely embraced by academics and professionals alike, to the extent that healthcare organizations in various countries are adjusting their strategies toward VBHC [6]. However, the implementation of these principles in real-world contexts has highlighted several challenges. The main difficulties encountered relate to the need for transformative leadership, clinical data collection and management, stakeholder engagement, and the fragmentation of healthcare services. In addition, evidence shows that healthcare organizations have move toward the VBHC approach by adopting only a few components of the VBHC agenda [6,7], with policymakers, hospital management, and healthcare providers pursuing separate strategies instead of a more strategic and coordinated approach. This disjointed approach makes it challenging to align efforts and achieve the comprehensive changes required. Indeed, although hospitals are at the core of this change, VBHC is a complex concept, and its implementation necessitates a range of actions and practices across multiple domains within the healthcare system, extending beyond the hospital’s boundaries [8,9,10]. Larson et al. [11] emphasize that a VBHC system requires the development of a governance and regulatory context. This necessitates the development of coordinated public policies, regulations, and shared infrastructure to encourage multi-stakeholder cooperation and value-based innovation across all healthcare sectors [11]. However, while the existing literature is primarily concerned with providing guidelines for implementing value-based care in hospital settings [6,12,13], a limited amount of research has explored how to create comprehensive and multilevel strategies for implementing VBHC, which are necessary for implementing the principles underlying this theory. Within this context, this study attempts to take a step forward. We aim to conduct a scoping review to identify studies that address the implementation of VBHC at different levels (healthcare policymakers, hospital management, and healthcare providers), focusing on each level’s relative strategy. This review addresses the following research questions: How can healthcare systems effectively transition to VBHC through strategic engagement, governance, and integrated care models at both the system and hospital levels? What managerial strategies should hospitals consider when transitioning to VBHC? These questions will guide our analysis and provide insights into the strategies for transitioning to VBHC.

## 2. Materials and Methods

We conducted and reported a scoping review in accordance with the Preferred Reporting Items for Systematic Reviews and Meta-Analyses extension for the scoping reviews (PRISMA-ScR) checklist [14] (see Appendix A). No published protocol is available for this scoping review. This scoping review was conducted using Arksey and O’Malley’s methodological framework [15], which consists of five stages: identifying the research question, identifying relevant studies, study selection, charting the data, collating, summarizing, and reporting the results. The decision to conduct a scoping review instead of a systematic review was made to identify the various types of evidence related to the implementation strategies of VBHC theory. To attain this objective, it is imperative to analyze papers that employ a range of diverse methodologies [16].

### 2.1. Search Strategy

The electronic databases of Web of Science, PubMed, MEDLINE, and Scopus, as well as the following keywords and synonyms, linked by Boolean operators, were used: “value-based care”, “VBHC”, “value-based”, “policy”, “strategy”, “intervention”, “organizational change”, “change management”, “redesign”, “transformation”, “implementation”. Search strategies for the four scientific electronic databases are provided in the online Appendix A. The investigation was conducted in September 2023, and the articles retrieved were screened according to the inclusion/exclusion criteria previously stated and listed in the following paragraph.

### 2.2. Study Eligibility

The inclusion criteria for this study encompassed both qualitative and quantitative studies published in English from January 2006, when Porter and Teisberg coined VBHC, to 31 July 2023 that describe the implementation of VBHC in a hospital setting or healthcare system, providing that they adhered to the following predefined criteria:They examined how government policies, network structures, and regulatory frameworks can support the transition to VBHC.They focused on hospital management strategies before implementation and throughout the process of adopting and sustaining VBHC practices.

We have excluded studies:Solely focusing on implementing individual components of VBHC.Describing different methodologies for designing and implementing bundled payments/value-based repayment program reimbursement.Reporting the evolution of VBHC theory.Not reporting on any managerial implementation strategies of VBHC theory.

### 2.3. Study Selection

The identified articles were uploaded to the Rayyan website for deduplication and screening. Two reviewers (dME and CA) independently assessed titles and abstracts for eligibility, resolving discrepancies through reviewers’ discussion or consulting a third party (DM). Articles meeting eligibility criteria underwent full-text reviews and data extraction. To ensure comprehensive coverage, reviewers also checked the reference lists of included studies.

Regarding the gray literature search, the reference lists of articles included in the scoping review were manually examined to identify additional relevant publications. A comprehensive Excel list of potentially eligible documents was compiled, and the full texts of these documents were subsequently assessed independently by two researchers (dME and DM) based on the inclusion criteria. Any discrepancies were resolved through discussion between the reviewers or, when necessary, by consulting a third party (DM).

### 2.4. Quality Appraisal

Based on the current methodological guidelines for scoping studies [14,16], no Critical Appraisal of the strength and quality of the included papers was performed.

This protocol was registered in Open Science Frame work (OSF; https://osf.io/6qfgx/, accessed on 31 October 2024) https://doi.org/10.17605/OSF.IO/N8P7Q.

### 2.5. Data Extraction, Analysis, and Reporting

From the included studies, two reviewers (dME, DM) extracted the following information on a dedicated data extraction form built with Excel:Study identification (first author, title, publication year).Study characteristics (country, study design).Level of analysis (system level, hospital level).Disease area (if relevant).Dimensions analyzed.Barriers and/or success factors.

## 3. Results

The screening and study selection is presented in Figure 1. The scientific and gray literature search produced 7790 records, which were then screened for duplicates, resulting in 3543 eligible records for title/abstract screening. Following this, the full text of 117 articles was carefully reviewed. Ultimately, 30 articles were included in the scoping review, consisting of a systematic literature review study (n = 1), a scoping review (n = 1), qualitative studies (n = 5), case studies (n = 12), mixed (explorative and qualitative) study designs (n = 3), a conceptual framework and case examples (n = 1), an observational cohort study design (n = 1), a comparative multiple case study (n = 2), explorative design studies (n = 3), and a descriptive qualitative study (n = 1). The articles selected for this research were published between 2013 and 2023 and originated from three countries, namely the Netherlands (n = 12), the United States (n = 9), and Sweden (n = 5).

### 3.1. Literature Synthesis

The data extraction is shown in Table 1. Subsequently, the data extraction results were translated into a framework (Table 2). This framework serves as a comprehensive roadmap to guide policymakers, healthcare providers, and hospital managers during their transition toward VBHC. It comprises four strategic levels: government commitment in policy definition, organizational vision and cultural integration, Operational Excellence, and VBHC assessment. Each strategic level is associated with specific dimensions and operational strategies that should be implemented.

### 3.2. Government Commitment in Policies Definition

Eight articles [18,19,21,22,24,27,33,43] described the role of government commitment in implementing the principles of the VBHC model, especially during the introduction of value-based payment reforms. These studies have identified several operational strategies. Four studies [18,22,24,43] suggested establishing state legislation to encourage state and regional initiatives for VBHC that would involve key stakeholders (such as policymakers, government agencies, public and private healthcare providers, and patients’ private organizations) in this change process [18,24]. Four studies [19,21,25,27] suggested that the government may need to provide further guidance or assistance for payment reform, adopt a long-term vision, provide information on implementation and the potential impact of payment reforms, as well as create a sense of urgency for the change [19,22,24,27,43]. Lastly, other studies have recommended allocating significant resources (financial/human resources) to support the transition, including external resources (e.g., grants) [22]. Regarding value-based agreements (VBAs), adopting legal and regulatory policies that enable innovative contracting, such as net price confidentiality, was recommended [21]. Policies that facilitate acquiring and using appropriate data to support contracting have also been suggested [21]. Finally, collaboration and engagement with the medical device industry is believed to be beneficial [43].

### 3.3. Organizational Vision and Cultural Integration

In the hospital setting, the need for an official commitment to value-based redesign from higher organizational levels was emphasized in seven studies [12,17,25,31,35,36,39]. This is consistent with the findings of another two included studies [20,34] suggesting that it is important to incorporate the implementation of the VBHC concept into hospital strategies, policy documents, planning, and control. Several papers suggested attributing formal responsibility and mandates to a steering group featuring hospital representatives for the implementation of VBHC [6,17,20,25,26,28,30,31,32,34,35,36,40]. Other studies suggested to empowering service line leadership with direct accountability and authority over programs and budgets [20,34]. Another frequently employed operational strategy entails developing tailored business plans to provide a structured, clear, and goal-oriented process, that is adaptable to each situation [10,25,34,35,40]. Furthermore, starting with “experiments” and “pilots” [12,23,26,35,40], planning and preparation before beginning the implementation process [36,39,40] was also regarded as essential in transitioning to VBHC. Five studies [6,23,35,36,40] described the support from consultancy since the early stages of the transition process as a success factor.

Once the vision and strategy are defined, the principles of VBHC need to be anchored in the hospital’s organizational culture. To achieve this, various operational strategies have been proposed in the literature. One commonly cited strategy was the education and training of clinical staff about the VBHC concept [6,12,20,25,32,34,36,37] in order to improve communication and information sharing with staff regarding the change [6,23,26,34,36,37,38] and to entail time for healthcare professionals to work on the project and anchor changes in their daily work [23,25,31,35,36]. Some studies underscored the importance of continued recognition of the usefulness of VBHC implementation [31,36,39] (with initially positive results [23,25,31,35]) and the motivation of individuals to become involved at each step of the developmental process [12,23,34]. Establishing and maintaining a positive atmosphere [34,35,39], trust building, and pursuing a common goal [17,34] were proposed multiple times. Finally, seven studies suggested involving patients and their representatives in the implementation process [10,25,31,36,38,40,42].

### 3.4. Operational Excellence

To improve regional networking models and promote cooperation between hospitals and local communities, four studies [10,27,28,30] stressed the importance of establishing transmural care standards. The term “transmural care standard” refers to a healthcare approach that integrates various care settings to ensure seamless patient management throughout the entire care delivery value chain. At the same time, four studies [18,27,33,37] highlighted the value of hiring additional staff dedicated to care coordination (such as nurses, navigators and community health workers) to expand care teams in the hospital setting. Participating in peer-to-peer learning collaboration on implementing new delivery models or enhancing care coordination [33] as well as planning and attending periodic networking meetings [27,30] were other strategies that were mentioned. In the hospital setting, process optimization and coordination can be achieved through the definition and optimization of Critical Pathways (CPs) [10,12,27,28,38,42], the utilization of liaison positions (such as ‘intermediary managers’) to enhance coordination between functional units [6,10,12,37,39], and the application of a lean methodology [25]. Regarding developing a multidisciplinary team, the following strategies were identified as effective means in the included studies in this study: engaging all professionals involved in the care of the same patient at different levels [10,20,35,39,42], planning and attending regular institutionalized meetings (‘standing committees’) [6,12,25], sharing workspaces [25], and creating multidisciplinary meetings to discuss complex patients [10,28]. The term “standing committees” refers to institutionalized meetings that take place regularly and enable interunit communication [12]. To improve the data collection throughout patients’ clinical pathways and to support clinical decision making, six studies recommended setting up innovative data-sharing mechanisms to provide real-time data to providers [18,19,21,22,27,43] or/and a care and information technology platform with the objective of facilitating communication between patients and healthcare professionals [23,25,36,37]. Furthermore, three articles recommended the creation of dashboards containing outcome measurements, PROMs/PREMs, and costs [12,29,31]. Finally, other articles emphasized the need for additional support staff, such as data analysts, project leaders, and care managers [10,25,36], to assist the healthcare workforce.

### 3.5. VBHC Assessment

Seventeen studies described clinical outcome measurements [10,12,23,25,27,28,29,30,31,32,34,36,37,38,40,41,42] as crucial components of VBHC. The main operational strategies employed to measure clinical outcomes include the identification and collection of relevant clinical outcome measurements [10,12,23,25,27,28,29,30,31,32,34,36,37,38,40,41], the mapping of the care processes for each respective group of patients [23,40], and the acquisition, processing, and dissemination of data in a time-efficient manner for internal reflection [12,29,31,32,34,41,42]. In order to enhance their comprehension from a clinical standpoint, two additional operational strategies were proposed in one included article. These strategies involve elucidating the rationale behind clinical outcome measurements in a more pedagogical manner and simplifying the PowerPoint presentations of measured outcomes [36]. Additionally, the articles highlighted the importance of benchmarking the outcome data between hospitals [12,25,29,32,40]. In addition to clinical outcome measurements, the collection of data on patient-reported measures (PREMs and PROMs) [25,27,28,31,38,41], the measurement of costs, and the implementation of an ‘Audit and Feedback’ (A and F) mechanism [28,30] were reported as part of the VBHC components.

## 4. Discussion

Our study aimed to conduct a scoping review to identify studies that address the implementation of VBHC at different levels (hospital and system variables), focusing on each relative defined strategy. This comprehensive perspective emphasizes the importance of multilevel strategies and stakeholder engagement, which provides a more systemic view of VBHC implementation beyond the single hospital focus. Findings are organized into four strategic levels, namely government commitment in policy definition, organizational vision and cultural integration, Operational Excellence, and VBHC assessment. Each strategic level is associated with specific dimensions and operational strategies that should be implemented.

Our results highlight how critical government involvement is in promoting and implementing VBHC. The studies reviewed have demonstrated various combinations of state legislation and reimbursement strategies. However, almost all have underscored three fundamental aspects: the importance of a strong governmental leadership, enhancing cooperation among key stakeholders involved in the healthcare delivery process, and the development of IT platforms to support service delivery and payment models. De Vries E.F. et al. [19] and Kissam SM et al. [33] emphasize the role of strong governmental leadership in stimulating value-based payment reforms. This can be achieved by providing a long-term vision, supporting knowledge development, and creating a sense of urgency for implementing payment reforms. In this context, establishing state/regional legislation to encourage VBHC initiatives needs to be considered [18,22,24,43]. Only a few countries have regulated this matter recently, with the United States (US) being one of the most notable examples. The Medicare Shared Savings Program, established by the Affordable Care Act in the US, promotes the formation of Accountable Care Organizations (ACOs) and several types of bundled payment models as part of public health system reform [44]. In the Netherlands, the extent of the government’s intervention to accelerate reform is unclear, as its role in payment reform is less defined [19,45]. Douglas C. et al. [18] reported that shifting towards value-based payments requires extensive cooperation among private healthcare plans, providers, purchasers, and public programs. However, this demands time and resources. The analysis of the papers included in the scoping review highlights several limits and barriers complicating the implementation of VBHC principles. In payment policy definition, the informational asymmetry among stakeholders poses a significant challenge to reforming payment models [19]. This issue is further exacerbated by the time required to establish an effective dialog among parties [21], who often have limited experience collaborating on VBAs [21]. Additionally, the fragmentation of EHR systems [22], often non-interoperable or incompatible, represents a significant barrier to data sharing among different care providers. This challenge is further compounded by numerous restrictions that limit the seamless and integrated exchange of patient information [33].

In the context of hospital organizations, our study indicates that it would be beneficial for the higher organizational levels to make official commitments to value-based redesign. Daniels et al. [25] suggested that a more formal commitment is needed to move the organization towards VBHC, and that implementation will only be successful if it is embedded in hospital strategy, policy documents, and planning and control. Preliminary experiments, pilot testing [12,23,27,35,40], and adequate planning and preparation before implementation [36,39,40] are also highly advisable. In their study, Zipfel et al. [35] identified personal and professional involvement in the design of the intervention as being an important role in successful implementation, as well as clinical leadership and a positive climate. However, our findings have identified several limits in regard to implementing VBHC principles within hospital settings. First, effective implementation requires a profound cultural shift [18], demanding a considerable investment of time and resources [18,24]. Moreover, evidence frequently highlights mutual distrust between clinicians and managers, often stemming from a lack of recognition of each other’s contributions [17,41]. This distrust risks diverting attention from the shared goal of maximizing patient value. Compounding these difficulties, traditional department-based structures organized by function further hinder the transition to a disease-centered organizational model [10,12,20,23,25,26]. To fill this gap, two distinct approaches to reshaping hospital organizational design for implementing VBHC principles from the papers included one that was more incremental and one that was more radical. As described by Steinmann et al. [12], the incremental hospital redesign approach employs linking mechanisms, such as middle managers and regular meetings of multidisciplinary teams, to facilitate coordination while preserving the original functional units. On the other hand, a radical approach was described by Ramos et al. [10] in which the traditional medical organization was replaced by a matrix organizational structure with vertical functions and horizontal medical themes, allowing patient flows to cross departmental boundaries. However, as reported in several studies, this shift from a traditional departmental structure to a disease-oriented organization is not a foregone conclusion. Several strategies that could overcome this gap emerged from our study [10,12,20,23,25], ranging from the definition of Critical Pathways, anchoring the VBHC principles in the hospital’s organizational culture through various initiatives, to planning regular institutionalized meetings (‘standing committees’).

However, in today’s healthcare environment, the vision is limited to considering the patients’ care as being exclusively hospital-based. This calls for the development of integrated care models which involve efforts to coordinate and join fragmented and disjointed healthcare providers [46]. Van Veghel H. P. A. et al. [30] stressed the importance of defining transmural care standards within the context of a Dutch regional networking to provide patients with care tailored to their individual needs. Our results suggest that cooperation can be enhanced by equipping healthcare providers with advanced health information technology (IT) systems and dashboards to capture the state of a patient across time and then sharing this information with various stakeholders who interact with the patient at different levels. However, as highlighted by several articles in our review [18,22,23,32,40], many healthcare providers lack necessary IT support. This structural deficit can hinder achieving another important goal of VBHC, which is to measure the outcomes and costs for each patient [2,3,8]. This led us to the fourth level of our analysis, known as “VBHC assessment”. This section covers all the main approaches and resources needed to improve the quality of care according to the VBHC theory. The most commonly cited dimension is the ”identification and collection of relevant clinical outcome measures” [10,12,23,25,27,28,29,30,31,32,34,36,37,38,40,41,42]. However, only a few papers have methodologically described the methods used to make this a routine part of hospital practice. The same applies to the “collection of patient-reported measures (PREMs and PROMs)”. With regard to cost measurement, although Time-Driven Activity-Based Costing (TDABC) is the predominant choice for cost accounting in this context [47,48], its practical adoption in hospitals has proven to be extremely challenging. Numerous studies highlight how measuring costs was particularly difficult to establish as the hospital accounting system only allowed for data capturing on an aggregated level [12,23]. Additionally, hospital care is still paid according to pay-for-volume contracts, and budget responsibility still lies with the traditional functional departments [10,25,26,36].

Our findings should be interpreted in the light of some limitations. Firstly, scoping reviews come with their own limitations. They typically do not assess the quality of the included studies as rigorously as systematic reviews do [14,16]. This can affect the reliability and validity of the findings, as lower-quality studies might be included without proper scrutiny. Additionally, scoping reviews may not always provide a comprehensive analysis of the evidence due to their broader focus on identifying and mapping the existing literature rather than synthesizing findings in an in depth manner [16]. Secondly, another limitation of this study is related to the eligibility criteria. Although payment models are a cornerstone of VBHC, papers describing different methodologies for designing and implementing value-based payments (VBPs) were not included because it is impractical to develop a uniform and scalable strategy globally given the diversity of healthcare systems in different countries. Thus, we have focused on papers that examine the governmental role as an active participant in this transition process. We decided to exclude from our review all articles describing the implementation of VBHC methodologies, such as the collection of Patient-Reported Experience Measures (PREMs) and Patient-Reported Outcome Measures (PROMs) or the adoption of TDABC, if such applications were related to isolated case studies and not integrated into daily hospital practice. In addition, the literature’s tendency to report only positive outcomes may have limited this review’s ability to capture critical or even unsuccessful aspects of VBHC implementation attempts in real-world settings. Lastly, we only included articles written in English, which may have introduced a country bias.

## 5. Conclusions

Our work provided an overview of the operational and management strategies adopted at different levels to implement VBHC principles in real-world settings. The study results underscored the necessity of a comprehensive approach to implement VBHC successfully. Indeed, while hospitals are recognized as critical players in this transition, a complete shift to VBHC necessitates government involvement in state legislation, reimbursement strategies, and hospital networking to ensure seamless patient management throughout the entire care delivery value chain.

In addition, the successful transition to VBHC necessitates a cultural transformation and the implementation of value-based quality improvement systems. This entails integrating clinical outcome monitoring, encompassing clinical outcomes and patient-reported measures (PROMs and PROMs), with audit and feedback (A and F) mechanisms to facilitate continuous improvement. Furthermore, developing innovative data-sharing systems and standardizing the measurement of outcomes is vital for enabling performance comparisons and supporting coordination across healthcare providers.

Finally, the review highlighted the limited number of studies evaluating the role of state committees in VBHC implementation, indicating a need for a more comprehensive examination of the state’s role in implementing VBHC.

## Figures and Tables

**Figure 1 healthcare-12-02457-f001:**
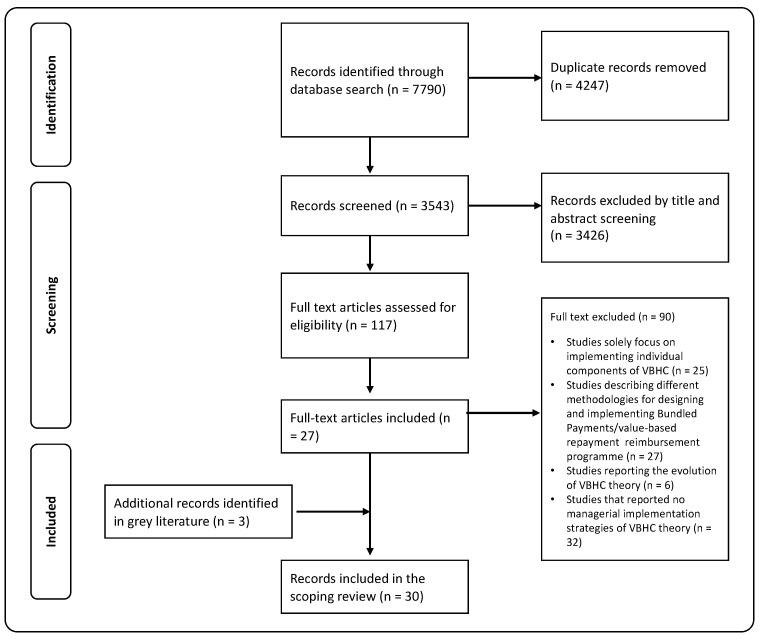
Flow chart of the selection and screening process in the scoping review.

**Table 1 healthcare-12-02457-t001:** Synthesis of the results.

First Author	Title	Year	Country	Study Design	Level of Analysis	Main Dimensions Analyzed	Barriers	Success Factors
Sze May Ng [17]	A qualitative study on relationships and perceptions between managers and clinicians and its effect on value-based healthcare within the national health service in the UK	2022	UK	A qualitative study	Hospital level	Vision and strategy Teamwork and interprofessional relationship	There was often mistrust between clinicians and managers, with both being dismissive of each other’s work.	Strong clinical leadership and medical engagement at all levels. Developed cultures where managers and clinicians are motivated and supported to work in partnerships.
Douglas Conrad [18]	A Report On Eight Early-Stage State And Regional Projects Testing Value-Based Payment	2013	US	A case study design	Government commitment	Value-based payment (VBP)	It requires time and resources in addition to culture change.	State legislation can galvanize regional and state payment reform initiatives. Previous experience in the development, implementation, and evaluation of VBP models
Eline F de Vries [19]	Barriers to payment reform: Experiences from nine Dutch population health management sites	2019	Netherlands	A qualitative study	Government commitment	Value-based payment (VBP)	Information asymmetry as a barrier towards payment reform. Hesitation to accept greater financial accountability.	//
Robert A. Phillips [20]	Creating and Maintaining a Successful Service Line in an Academic Medical Center at the Dawn of Value-Based Care: Lessons Learned From the Heart and Vascular Service Line at UMass Memorial Healthcare	2015	US	A case study design	Hospital level	Teamwork and interprofessional relationship	The traditional departmental structure made it difficult to fully shift towards disease-oriented organization.	Dedicated leadership with accountability and responsibility for budget.
Elizabeth A. Griffiths [21]	Demonstrating proof of concept for value-based agreements in Europe: two real-world cases	2023	UK	A case study design	Government commitment	Value-based agreements (VBAs)	The limited experience of both parties in working together on VBAs. It took time to develop a dialog between the relevant stakeholders and to determine partners willing to collaborate on the scheme design.	Engagement of core stakeholders (payers, manufacturers, and physicians) was critical.
Douglas Conrad [22]	Emerging Lessons From Regional and State Innovation in Value-Based Payment Reform: Balancing Collaboration and Disruptive Innovation	2014	UK	A qualitative study	Government commitment	Value-based payment (VBP) Development and implementation health IT infrastructure	The prevalence of disparate EHR systems that are not interoperable or cannot adapt to one another.	Previous experience in the development, implementation, and evaluation of VBP models. Pressure from policymakers, regulators, and organized public and private purchasers.
Kerstin Nilsson [23]	Experiences from implementing value-based healthcare at a Swedish University Hospital—a longitudinal interview study	2017	Sweden	A mixed (explorative and qualitative) design	Hospital level	Anchoring the new approach in the hospital organizational culture Cost measurement	The lack of IT systems supporting VBHC. The traditional departmental structure made it difficult to fully shift towards disease-oriented organization. Measure costs were particularly difficult to establish as the hospital accounting system only allowed for data capturing on an aggregated level.	Leadership skills, such as communication and motivation, to get people involved step by step in developing the process.
Diogo LL Leao [24]	Facilitating and Inhibiting Factors in the Design, Implementation, and Applicability of Value-Based Payment Models: A Systematic Literature Review	2023	Netherlands	A Systematic Literature Review	Government commitment	Value-based payment (VBP)	It requires time and resources.	Previous experience in the development, implementation, and evaluation of VBP models. High motivation, engagement, and trust among involved stakeholders. Transparency and communication among involved stakeholders.
Kirsten Daniels [25]	Five years’ experience with value-based quality improvement teams: the key factors to a successful implementation in hospital care	2022	Netherlands	A qualitative study	Hospital level	Vision and strategy Teamwork and interprofessional relationship Cost measurement	The traditional departmental structure made it difficult to fully shift towards disease-oriented organization. Hospital care is still paid according to pay-for-volume contracts, and budget responsibility still lies with the traditional functional departments.	To have one team that addresses all quality-improvement-related topics.
Luc Theunissen [26]	Implementing Value-Based healthcare Principles in the Full Cycle of Care: The Pragmatic Evolution of the Netherlands Heart Network	2023	Netherlands	A case study design	Regional integrated care system	Teamwork and interprofessional relationship Cost measurement	The traditional departmental structure made it difficult to fully shift towards disease-oriented organization. Hospital care is still paid according to pay-for-volume contracts, and budget responsibility still lies with the traditional functional departments.	Platform where organizations can connect to share data and best practices.
Douglas Conrad [27]	Implementing Value-Based Payment Reform: A Conceptual Framework and Case Examples	2015	US	A Conceptual Framework and Case Examples	Government commitment	Value-based payment (VBP)	//	Clear and consistent communication about movement toward larger reforms.
Dennis van Veghel [28]	Improving clinical outcomes and patient satisfaction among patients with coronary artery disease: an example of enhancing regional integration between a cardiac center and a referring hospital	2020	Netherlands	An observational cohort study design	Regional integrated care system	Integrated care	//	Trust and cooperation with other institutes.
P. B. van der Nat [29]	Insights on value-based healthcare implementation from Dutch heart care	2020	Netherlands	A case study design	Regional integrated care system	Integrated care	//	Trust and cooperation with other institutes.A platform where organizations can connect to share data and best practices.
H. P. A. van Veghel [30]	Introducing a method for implementing value-based healthcare principles in the full cycle of care: Using atrial fibrillation as a proof of concept	2020	Netherlands	A case study design	Regional integrated care system	Integrated care	//	Platform where organizations can connect to share data and best practices.
Pedro Ramos [10]	It takes two to dance the VBHC tango: A multiple case study of the adoption of value-based strategies in Sweden and Brazil	2021	Sweden and Brasil	A comparative multiple case study	Hospital level	Teamwork and interprofessional relationship Cost measurement	The traditional departmental structure made it difficult to fully shift towards disease-oriented organization. The financing of care was not aligned with the care production and outcomes monitoring.	The involvement of clinical staff was a crucial factor.
Dane Lansdaal [31]	Lessons learned on the experienced facilitators and barriers of implementing a tailored VBHC model in a Dutch university hospital from a perspective of physicians and nurses	2021	Netherlands	A descriptive qualitative study	Hospital level	Vision and strategy Support of information technology	The usage of the EHR in daily practice.	Continued recognition of the usefulness of the VBHC implementation.
Dennis van Veghel [32]	Organization of outcome-based quality improvement in Dutch heart centers	2020	Netherlands	A mixed-method approach	Regional integrated care system	Quality improvement initiatives (QI)	Insufficient data infrastructure for successful outcome-based quality improvement.	//
Gijs Steinmann [12]	Redesigning value-based hospital structures: a qualitative study on value-based healthcare in the Netherlands	2022	Netherlands	A qualitative exploratory study	Hospital level	Teamwork and interprofessional relationship Cost measurement	The traditional departmental structure made difficult to fully shift towards disease-oriented organization. Measuring costs was particularly difficult as the hospital accounting system only allowed for data collection on an aggregated level.	To have multiple leaders, each representing a particular organizational component.
Kissam SM [33]	States Encouraging Value-Based Payment: Lessons From CMS’s State Innovation Models Initiative	2019	US	A qualitative study	Government commitment	Value-based payment (VBP)	Lack of multipayer alignment around VBP models. Restrictions on the ability to share patient data across all care providers	//
J. Seth Chatfield [34]	Ten CEO Imperatives for Healthcare Transformation: Lessons From Top-Performing Academic Medical Centers	2017	US	A mixed-method approach	Hospital level	Vision and strategy anchoring the new approach in the hospital organizational culture	//	To have a common shared vision and align goals at all the levels with that vision. Effective communication.
Nina Zipfel [35]	The implementation of change model adds value to value-based healthcare: a qualitative study	2019	Netherlands	A case study design	Hospital level	Vision and strategy anchoring the new approach in the hospital organizational culture	An implementation approach was lacking to guide improvement interventions.	Multi-stakeholders’ involvement in the design of the intervention played an important role in the success of the implementation.
Dorine J. van Staalduinen [6]	The implementation of value-based healthcare: a scoping review	2022	Netherlands	A scoping review	Hospital level	Anchoring the new approach in the hospital organizational culture	//	Creating and enhancing leadership was also considered essential in transforming to VBHC.
Kerstin Nilsson [36]	The need to succeed—learning experiences resulting from the implementation of value-based healthcare	2018	Sweden	An explorative design	Hospital level	Vision and strategy cost measurement	Measuring costs was particularly difficult as the hospital accounting system only allowed data collection on an aggregated level.	The importance of planning and preparation before starting the implementation process.
Geralyn Randazzo [37]	Transitioning From Volume to Value. A Strategic Approach to Design and Implementation	2016	US	A case study design	Regional integrated care system	Integrated care and support of information technology	//	Platform where organizations can connect to share data and best practices. The Care Navigator role to assist the patients during their transition from the inpatient setting back to the community.
Giulia Goretti [38]	Value-Based Healthcare and Enhanced Recovery After Surgery Implementation in a High-Volume Bariatric Center in Italy	2020	Italy	A case study design	Hospital level	Teamwork and interprofessional relationship	//	Engaging patients and their representatives during the implementation work.
Kerstin Nilsson [39]	Value-based healthcare as a trigger for improvement initiatives	2017	Sweden	An explorative design study	Hospital level	Teamwork and interprofessional relationship	//	To create an open and trusting communication environment to succeed with developing processes.
Christian Colldén [40]	Value-based healthcare translated: a complementary view of implementation	2018	Sweden	A case study design	Hospital level	Support of information technology	The lack of IT systems supporting VBHC.	//
Chancellor F. Gray [41]	Value-based Healthcare: “physician Activation”: Healthcare Transformation Requires Physician Engagement and Leadership	2020	US		Hospital level	Teamwork and interprofessional relationship	Communication barriers between the physician and nonphysician health system leaders.	//
Aakash Keswani [42]	Value-based Healthcare: Part 1-Designing and Implementing Integrated Practice Units for the Management of Musculoskeletal Disease	2016	US	A case study design	Hospital level	Teamwork and interprofessional relationship	//	Engaging and activating patients throughout the care cycle, incorporating patients’ goals and preferences in treatment decisions (via SDM). Strong clinical leadership to promote teamwork, collaboration, and joint accountability for patient outcomes and overall cost of care.
Joon Hurh [43]	Value-based healthcare: prerequisites and suggestions for full-fledged implementation in the Republic of Korea	2017	Republic of Korea	A case study design	Government commitment	Value-based payment (VBP)	//	Commitment and support from healthcare providers by normalizing payment rates for healthcare providers.

**Table 2 healthcare-12-02457-t002:** VBHC Implementation: Strategy and Operational Dimensions.

Strategic Level	Dimension	Operational Strategies	References
Government commitment in policies definition	Value-based payment (VBP)	Providing more guidance or assistance in payment reforms through a long-term vision with information on the implementation and potential impact of payment reforms.	[19,21,25,27]
		Creating a sense of urgency for implementing payment reforms.	[19,22,24,27,43]
		Defining national or local legislation.	[18,22,24,43]
		Involving key stakeholders in the change process and implementation of the models.	[18,24]
		Allocating significant resources toward payment and delivery system innovations.	[22]
	Value-based agreements (VBAs)	Legal/regulatory policies permitting innovative contracting (e.g., net price confidentiality).	[21]
		Policies supporting appropriate data capture and use to support contracting.	[21]
		Collaborating and engaging with the medical device industry.	[43]
Organizational vision and cultural integration	Vision, strategy and governance structures	Having an official commitment to value-based redesign from the higher levels of the organization.	[12,17,25,31,35,36,39]
		Embedding the adoption of the VBHC concept in the hospital strategy, policy documents, and planning and control.	[20,34]
		Providing formal responsibility and mandates to a steering group with hospital representatives for the implementation of VBHC.	[6,17,20,25,26,28,30,31,32,34,35,36,40]
		Empowerment of service line leadership with direct accountability and authority over programs and budgets.	[20,34]
		Developing a tailored business plan to provide a structured process that is clear, goal-oriented, and adaptable to each situation.	[10,25,34,35,40]
		Being supported by consultancies.	[6,23,35,36,40]
		Starting with “experiments” and “pilots”.	[12,23,26,35,40]
		Planning and preparation before starting the implementation process.	[36,39,40]
	Anchoring the new approach to the hospital organizational culture	Staff training and education on the VBHC concept.	[6,12,20,25,32,34,36,37]
		Improving communication and information with staff about the change.	[6,23,26,34,36,37,38]
		Providing time for healthcare professionals to work on the project and anchoring changes to their daily work.	[23,25,31,35,36]
		Continued recognition of the usefulness of the VBHC implementation.	[31,36,39]
		Starting with positive results.	[23,25,31,35]
		Motivating people to get them involved step by step in developing the process.	[12,23,34]
		Involving patients and their representatives in the implementation process.	[10,25,31,36,38,40,42]
Operational Excellence	Standardize care pathways	Defining transmural care standard.	[10,27,28,30]
		Hiring additional staff dedicated to care coordination to connect the territory and the hospital.	[18,27,33,37]
		Participating in peer-to-peer learning collaboratives on implementing new delivery models or enhancing care coordination.	[33]
		Planning and attendance of periodical networking meetings.	[27,30]
		Defining and optimizing Critical Pathways (CPs).	[10,12,27,28,38,42]
		Using liaison positions (such as “intermediary managers”) to enhance coordination between functional units.	[6,10,12,37,39]
		Appling the lean-methodology.	[25]
	Developing multidisciplinary teams	Engaging all professionals involved in the different levels of one patient’s care	[10,20,35,39,42]
		Planning and attendance of regularly institutionalized meetings (“standing committees”).	[6,12,25]
		Sharing workspace.	[25]
		Creating multidisciplinary meetings to discuss complex patients.	[10,28]
	IT support	Setting up innovative data sharing mechanisms to provide real time data to providers.	[18,19,21,22,27,43]
		Setting up care and information technology platforms to facilitate both patients and healthcare professionals.	[23,25,36,37]
		Creating dashboards containing outcome measurements, PROMs/PREMs and costs.	[12,29,31]
	Additional resources	Availability of additional support staff (data analysts/project leaders/care managers).	[10,25,36]
VBHC assessment	Clinical Outcome measurement	Identifying and collecting relevant clinical outcome measurements.	[10,12,23,25,27,28,29,30,31,32,34,36,37,38,40,41]
		Mapping the care processes for each respective group of patients.	[23,40]
		Benchmarking outcome data among hospitals.	[12,25,29,32,40]
		Obtaining, processing, and dispersing data in a time-efficient manner for internal reflection.	[12,29,31,32,34,41,42]
		Explaining the clinical outcome measurements more pedagogically.	[36]
		Simplifying PowerPoint presentations of outcomes measured.	[36]
	Patient-reported measures	Collecting data regarding patient reported measures (PROMs and PREMs).	[25,27,28,31,38,41]
	Costs measurement	Measuring costs based on actual resource use over the full cycle of care for the patient’s condition.	
	Audit and Feedback (A and F)	Performing “Audit and Feedback” (A and F).	[28,30]

## Data Availability

Not applicable.

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
