# Peer review of "Moving from Principles to Practice: A Scoping Review of Value-Based Healthcare (VBHC) Implementation Strategies"

_healthcare, 2024, doi:10.3390/healthcare12232457_

Round 1

Reviewer 1 Report

Comments and Suggestions for Authors

Review Form

Comments and suggestions for authors

The manuscript with the title “Moving from Principles to Practice: A Scoping Review of Value- Based Healthcare (VBHC) Implementation Strategies” addresses a very important topic.

My comments about the article content are the following ones:

Introduction: well organized and well founded and emphasizes the need to study this topic.

In order to provide a complete picture, it would be useful to add the aspect of patients' perception of health, which differs from that of physicians especially with regard to psychological well-being. It should be emphasized that good patient management also includes this aspect. It is important to act not only to implement treatment strategies based on clinical practices, but also those that aim to improve empathy between doctors and patients in order to achieve better compliance. The following article could be added to add these considerations:

-          Fineschi D, Acciai S, Napolitani M, Scarafuggi G, Messina G, Guarducci G, Nante N. Game of Mirrors: Health Profiles in Patient and Physician Perceptions. Int J Environ Res Public Health. 2022 Jan 21;19(3):1201.

Materials and Methods: it would be appropriate to implement the search strategy section with the search strings used to select articles for inclusion in the review.

Results: three articles from the grey literature are reported, it would be appropriate to look into this aspect in more detail.

Discussion: summarizes the findings of the review, including the limitations section.

Conclusions: provides a general interpretation of the results and their implication for practice.

Author Response

Dear Reviewer,

Thank you for your valuable suggestions. Below, I will address each point (n red) and explain how I have improved the paper based on your feedback.

Introduction: well organized and well founded and emphasizes the need to study this topic.

In order to provide a complete picture, it would be useful to add the aspect of patients' perception of health, which differs from that of physicians especially with regard to psychological well-being. It should be emphasized that good patient management also includes this aspect. It is important to act not only to implement treatment strategies based on clinical practices, but also those that aim to improve empathy between doctors and patients in order to achieve better compliance. The following article could be added to add these considerations:

-          Fineschi D, Acciai S, Napolitani M, Scarafuggi G, Messina G, Guarducci G, Nante N. Game of Mirrors: Health Profiles in Patient and Physician Perceptions. Int J Environ Res Public Health. 2022 Jan 21;19(3):1201.

Thank you for your kind suggestion. We have revised it as follows (Pag. 1-2, lines 41-46)

“This approach puts the patient at the centre of the care process, facilitating their active involvement in decisions concerning their health through the Shared Decision Making (SDM) approach. This method integrates clinical expertise with the patient's values, needs, and preferences, fostering an empathetic and collaborative dynamic that strengthens the physician-patient relationship, yielding benefits for both parties.”

Materials and Methods: it would be appropriate to implement the search strategy section with the search strings used to select articles for inclusion in the review -> Thank you for highlighting the need for clarification. We included the search strings in the online Supplemental File 2, leaving only a selection of keywords and synonyms used in those search strings within the Methods section.

Results: three articles from the grey literature are reported, it would be appropriate to look into this aspect in more detail -> Thank you for highlighting the need for clarification. In analyzing the papers in our scoping review, we did not distinguish between those identified through the search strategy and those retrieved from grey literature. Accordingly, the analysis was conducted collectively, encompassing all included articles in a unified manner.

Discussion: summarizes the findings of the review, including the limitations section -> Thank you for your kind suggestion. In the Discussion section, we examined the main findings of our scoping review, providing concrete examples from the included papers and comparing and contrasting them with previous studies. As for the limitations of the paper, they have been included on page 12, lines 399-419.

Conclusions: provides a general interpretation of the results and their implication for practice. -> Thank you for your kind suggestion. We have revised it as follows (Pag. 12-13, lines 421-437):

“Our work provided an overview of operational and management strategies adopted at different levels to implement VBHC principles in real-world settings. The study results underscored the necessity of a comprehensive approach to implement VBHC successfully. Indeed, while hospitals are recognized as critical players in this transition, a complete shift to VBHC necessitates government involvement in state legislation, reimbursement strategies, and hospital networking to ensure seamless patient management throughout the entire care delivery value chain.

In addition, the successful transition to VBHC necessitates a cultural transformation and the implementation of value-based quality improvement systems. This entails integrating clinical outcome monitoring, encompassing clinical outcomes and patient-reported measures (PROMs & PROMs), with audit and feedback (A&F) mechanisms to facilitate continuous improvement. Furthermore, developing innovative data-sharing systems and standardizing the measurement of outcomes is vital for enabling performance comparisons and supporting coordination across healthcare providers.

Finally, the review highlighted the limited number of studies evaluating state committees' role in VBHC implementation, indicating a need for a more comprehensive examination of the state's role in implementing VBHC.”

Reviewer 2 Report

Comments and Suggestions for Authors

Value-Based Healthcare (VBHC) is an important and interesting topic that should be considered in the sustainability of healthcare systems and in improving the performance of healthcare organizations. Despite the theoretical recognition of the VBHC, the implementation difficulties have been the subject of multiple discussions. Although no country has fully adopted the VBHC agenda, it becomes evident that different theoretical framework elements operate better in some healthcare systems than others. Furthermore, some critics express reservations about the VBHC guide's assertions about being the appropriate path forward, reinforced by diverse understandings of what value for patients entails and the lack of evidence relating to VBHC's effectiveness. Other authors also argue that a single-minded focus on VBHC may cause serious infringements on medical ethical principles.

Reading the introduction (1), I felt that only positive aspects are considered, and VBHC is like a panacea for solving problems in health systems and health organizations. So, from my perspective, some positive and negative aspects should be balanced in the introduction section.

In the materials and methods section (2), it is not clear what the sources of grey literature are. Were some results from electronic databases used in the study, or were other databases specialized in grey literature also considered? Also, what kind of decision was implemented to solve discrepancies between reviewers? Did the reviewers use any agreement coefficient? It is also mentioned that "no published protocol is available for this scoping review, " but did the authors register the protocol in some service like PROSPERO?

The section of Results (3) has an error in numbering sections (starts with 4.1). This section and discussion are in line with the materials and methods section. However, in the literature synthesis, it is unclear what evidence or reasons support considering the four strategic levels of government commitment in policy definition, organizational vision and cultural integration, operational excellence, and VBHC assessment. These strategic levels emerged from the materials analysis, or are they the consequence of some analysis framework?

At the end of the document, it is essential to mention that the protocol was not published (or registered?), and no appraisal of the strength and quality of the included papers was performed.

Author Response

Dear Reviewer,

Thank you for your valuable suggestions. Below, I will address each point (in red) and explain how I have improved the paper based on your feedback.

Value-Based Healthcare (VBHC) is an important and interesting topic that should be considered in the sustainability of healthcare systems and in improving the performance of healthcare organizations. Despite the theoretical recognition of the VBHC, the implementation difficulties have been the subject of multiple discussions. Although no country has fully adopted the VBHC agenda, it becomes evident that different theoretical framework elements operate better in some healthcare systems than others. Furthermore, some critics express reservations about the VBHC guide's assertions about being the appropriate path forward, reinforced by diverse understandings of what value for patients entails and the lack of evidence relating to VBHC's effectiveness. Other authors also argue that a single-minded focus on VBHC may cause serious infringements on medical ethical principles.

Reading the introduction (1), I felt that only positive aspects are considered, and VBHC is like a panacea for solving problems in health systems and health organizations. So, from my perspective, some positive and negative aspects should be balanced in the introduction section. 

-> Following your insightful suggestion, we have integrated some negative aspects into various sections of the paper, specifically:

  • Introduction: Page 1, lines 50–58
  • Discussion: Page 10, lines 328–337 and 347–355 and Page 11, lines 382–383, 396–398, and 415–418

In the materials and methods section (2), it is not clear what the sources of grey literature are. Were some results from electronic databases used in the study, or were other databases specialized in grey literature also considered? Also, what kind of decision was implemented to solve discrepancies between reviewers? Did the reviewers use any agreement coefficient?

Thank you for your kind suggestion. As stated on page 3, lines 138–143, regarding the grey literature search, we did not conduct an analysis using databases. However, we studied the reference lists of the included articles. Perhaps the current phrasing was not entirely clear, and thanks to your suggestion, we have revised it as follows:

“Regarding the grey literature search, reference lists of articles included in the scoping review were manually checked and searched for additional relevant publications. An Excel list containing potentially eligible documents was created, and subsequently, the full texts of these documents were independently assessed by two researchers (dME and DM) against our inclusion criteria, resolving discrepancies through reviewers’ discussion or consulting a third party (DM)”

As regard the use of an agreement coefficient, we did not use them.

It is also mentioned that "no published protocol is available for this scoping review, " but did the authors register the protocol in some service like PROSPERO?

Thank you for your kind suggestion. Initially, we did not register the protocol on PROSPERO, as its website specifies that scoping reviews are not eligible for inclusion (https://www.crd.york.ac.uk/prospero/#aboutregpage). However, following the submission of our paper to this journal, and upon the recommendation of Ms. Maria Blanaru, Assistant Editor, we registered the protocol in the Open Science Framework database (https://osf.io/).

Here is the Open Science Framework database registration number:  https://osf.io/6qfgx/

The section of Results (3) has an error in numbering sections (starts with 4.1).

Thank you for your kind suggestion. We made these corrections immediately.

This section and discussion are in line with the materials and methods section. However, in the literature synthesis, it is unclear what evidence or reasons support considering the four strategic levels of government commitment in policy definition, organizational vision and cultural integration, operational excellence, and VBHC assessment. These strategic levels emerged from the materials analysis, or are they the consequence of some analysis framework?

Thank you for highlighting the need for clarification. These strategic levels were not based on a pre-existing framework. Identifying the four strategic levels is closely tied to the objective of our scoping review, which was "to identify studies that address the implementation of VBHC at different levels (healthcare policymakers, hospital management, and healthcare providers), focusing on each level's relative strategy."

Specifically, these levels emerged as a result of synthesizing the evidence found in our scoping, where the strategies implemented at each level were categorized into overarching themes:

  1. Government commitment in policy definition reflects the critical role of policymakers in creating an enabling environment for VBHC, as emphasized in studies addressing national frameworks and policy alignment.
  2. Organizational vision and cultural integration pertain to strategies employed at the hospital management level to foster a shared vision and integrate VBHC principles into the organizational culture.
  3. Operational excellence highlights efforts at the provider level to enhance care delivery processes through data optimization and efficiency improvements.
  4. VBHC assessment represents strategies for evaluating outcomes and ensuring alignment with VBHC goals across all levels.

At the end of the document, it is essential to mention that the protocol was not published (or registered?), and no appraisal of the strength and quality of the included papers was performed. 

Thank you for your kind suggestion. We insert these elements at page 3, lines 148-149.